# Bioinspired Hydrogels as Platforms for Life-Science Applications: Challenges and Opportunities

**DOI:** 10.3390/polym14122365

**Published:** 2022-06-11

**Authors:** Maria Bercea

**Affiliations:** “Petru Poni” Institute of Macromolecular Chemistry, 700487 Iasi, Romania; bercea@icmpp.ro

**Keywords:** hydrogel, bioinspired structure, tunable properties, self-healing, biological properties, mechanical properties, biomacromolecules, targeted applications

## Abstract

Hydrogels, as interconnected networks (polymer mesh; physically, chemically, or dynamic crosslinked networks) incorporating a high amount of water, present structural characteristics similar to soft natural tissue. They enable the diffusion of different molecules (ions, drugs, and grow factors) and have the ability to take over the action of external factors. Their nature provides a wide variety of raw materials and inspiration for functional soft matter obtained by complex mechanisms and hierarchical self-assembly. Over the last decade, many studies focused on developing innovative and high-performance materials, with new or improved functions, by mimicking biological structures at different length scales. Hydrogels with natural or synthetic origin can be engineered as bulk materials, micro- or nanoparticles, patches, membranes, supramolecular pathways, bio-inks, etc. The specific features of hydrogels make them suitable for a wide variety of applications, including tissue engineering scaffolds (repair/regeneration), wound healing, drug delivery carriers, bio-inks, soft robotics, sensors, actuators, catalysis, food safety, and hygiene products. This review is focused on recent advances in the field of bioinspired hydrogels that can serve as platforms for life-science applications. A brief outlook on the actual trends and future directions is also presented.

## 1. Introduction

The emerging technologies that pave the way for science and research development, and their impact in the economic activities, as well as human health and welfare, require more and more complex hydrogels with multiple functions. The actual challenge of the research is to decide, for a given material, what kind of functional properties should be taken into account with priority, and what kind of gelling strategy it needs. The originality of this review is the focus on evidencing the structural diversity and summarizing fabrication methods and properties of bioinspired hydrogels based on recent published information. Learning from nature, hierarchical self-assembly of biological compounds and their derivatives, very often in combination with various synthetic molecules, into more and more sophisticated structures, are continuously reported.

Hydrogels, some of the most investigated materials, are three-dimensional hydrophilic networks formed by physical, chemical, or both types of interactions between natural or synthetic macromolecules. Depending on their nature, crosslinking density, and hydrophilicity of the segments between two crosslinking points, such versatile networks can absorb a high amount of water. From a chemical point of view, the hydrogels can be obtained from natural or synthetic water soluble (co)polymers of various structures and architectures, interpenetrated polymer networks (IPNs), proteins, peptides, clays, multicomponent systems, etc., having different morphologies and functions [1,2,3]. In particular, smart networks able to respond to physical, chemical, and biological stimuli gained much attention for a wide range of applications: tissue engineering [4], bone regeneration [5], controlled-release drug delivery vehicles [6], wound healing [7], soft robotics [8], biosensing [9], intelligent electronics and artificial intelligence [10], actuators [11,12,13], stretched electronic devices [14], hygiene products [15,16], contact lens [17], cosmetics [18], food nutrition and health, food safety and food engineering and processing [19], advanced wastewater treatment [20], catalysis [21,22,23], etc.

Inspired by nature, the great challenge is now to create self-healing hydrogels that present the ability to form spontaneously new physical or chemical bonds when some existing bonds are disrupted, preserving the structural integrity and extending the service lifetime in various applications [24,25,26,27,28,29].

Hydrogels’ performances can be analyzed through their biological and mechanical properties, biodegradability, porosity, and swelling ability. These aspects will be underlined in the next sections.

## 2. Hydrogel Design as a Function of the Targeted Application

In the body, the biological network-like structures, such as muscles, cartilages, tendons, or heart valves, possess excellent mechanical properties, being very tough, strong, resilient, adhesive, and fatigue-resistant [30]. These biological hydrogels provide inspiration for the researchers working in the design of hydrogels with tunable mechanical properties, that are important for many biomedical applications which improve people’s health and daily life (such as: tissue engineering, medical implants, contact lenses, wound dressings, drug delivery, sensors and actuators, electronic devices, and soft robots). The map of mechanical properties distribution in the body is discussed by Zhao et al. [30] and it is briefly presented below: tensile strength in aorta: around 4 MPa; fracture toughness in heart valve: around 1 kJ/m^2^; in tendon: tensile strength of about 10 MPa, fracture toughness of approx. 30 kJ/m^2^; fatigue threshold of approx. 1 kJ/m^2^; for skeletal muscle: fracture toughness around 2.5 kJ/m^2^; fatigue threshold around 1 kJ/m^2^; articular cartilage has fracture toughness of approx. 1.8 kJ/m^2^; and the tendon, cartilage, ligament–bone interfaces present interfacial fatigue threshold around 0.8 kJ/m^2^.

There were identified several general principles that guide the design of hydrogels: (a) the use of one or more approaches to obtain networks with improved mechanical and physical properties, (b) the implementation of new strategies using unconventional polymer networks, and (c) orthogonal design of hydrogels to achieve multiple combined physico-chemical, mechanical, and biological properties [30,31,32,33]. By a careful synthesis and in some cases by combining various approaches (physical, chemical, enzymatic, irradiation), biocompatible hydrogels can be obtained, with antimicrobial and biodegradable properties and mechanical characteristics adjusted to suit the targeted applications. There is a variety of strategies to develop materials with properties attractive in many biomedical applications. Thus, the hydrogel strength can be tuned from their formulation, from very soft to very hard structure networks, suitable for the targeted applications: from injectable hydrogels [26,34,35] for tissue filling and restoration [3,36,37,38], biomaterials for rapid hemostasis [39], wound healing [39,40], and drug delivery carriers [35,41] to robust and tough hydrogels required for load-bearing applications (artificial cartilage, muscle) [34] and smart biomaterials for tissue engineering [4,24,42] or advanced technological applications [14,43,44].

The hydrogels properties should be compatible with those of the living systems. The mechanical strength of hydrogels can be controlled by the nature of network and it influences the adhesion, diffusion, and differentiation processes. The elastic modulus strongly depends on the type of structure (crosslinks, association, entanglements) and interactions present in the network. According to the literature data, mechanical strengths between 0.1 kPa–100 kPa are required in applications such as brain, neural cell or bone cell encapsulation and muscles [45,46,47] or higher than 80 MPa for the packaging materials [48].

Self-assembled hydrogels based on polysaccharide derivatives and synthetic polymers are used in cartilage tissue engineering because of superior mechanical properties and swelling ability [27,49,50] and, due to their 3D networks, the spherical morphology of encapsulated cells is preserved [50]. However, the physical hydrogels do not meet all requirements, some properties of interest need to be improved for biomedical applications. In such cases, suitable crosslinking agents are used [51,52], in particular for enhancing the mechanical strength and prolonging the duration of degradation [53] or as fillers of defects in the case of in vivo applications [54].

In order to improve hydrogel properties and functionalities, combined physical and chemical crosslinking can be used in interpenetrating polymer networks or they can be reinforced by interlocking polymers within the entangled networks, by ionic interactions or host–guest complexation [55,56]. These aspects will be discussed in the next sections, with an emphasis on bioinspired procedures.

### 2.1. Bioinspired Approaches for the Design of Hydrogels with Targeted Properties

In nature, the amino acids, nucleotides, and monosaccharides form macromolecular structures, such as: proteins, DNA, and polysaccharides, respectively. Macromolecules of different structures and architectures can be synthesized using a diversity of available monomers. By mimicking the nature at different levels, various hydrogel structures can be now developed focusing on the targeted properties, such as tunable mechanical properties or stimuli-responsiveness, variable porosity, or improved interactions with cell [3,10,11,12,39,42,44,47,55,56]. Thus, the research directions concerning the hydrogels are extended to diverse application fields as well. Biomolecules present physiochemical and biological characteristics that are suitable for biomaterials, such as: biocompatibility, antimicrobial properties biodegradability, and non-toxicity. However, in contrast to structures found in biological soft tissues, synthetic hydrogels have limited biological functions [3,34,55,56,57,58].

The major challenges in the hydrogel design are oriented to mimic the biological structures at different length scales and to reproduce different functions found in natural systems in order to increase the quality of life through repairing the damaged tissues and restoring functions, maintaining the good health state, ensuring food security, access to high level technologies, guaranteeing at the same time both environmental protection and society progress through deep knowledge of nature principles. Two major directions can be detected in the current approaches:-Synthesis of new molecules (monomers, functionalized peptides) that are then assembled in 3D networks;-Use of network structures (natural or synthetic molecules) in various combinations to give them new features and functionalities.

The progress registered in chemical synthesis, in parallel to materials engineering, contributed to the improvement of the mechanical and functional properties of hydrogels, through the new approaches, such as sliding cross-linking agents (guest−host complexes), interpenetrating networks, self-assembly peptides or block and graft copolymers, the control of hydrophile/hydrophobe balance, the design of microgels or nanocomposite hydrogels.

*Gelation and network formation* can be induced in different ways, either through chemical, physical, or combined crosslinking, as it was discussed in recent comprehensive reviews [1,2,4,18,19,20,30,55,56,57,58,59,60]. The characteristics of the building blocks influence in high extent the physico-chemical, mechanical, and biological properties of the hydrogels. The permanent covalent crosslinking uses various chemically synthesized and biological crosslinkers, condensation or thiol-click reactions, high energy irradiation, etc. [50,51,52]. However, compared to biological crosslinkers, chemically synthesized crosslinking agents have some disadvantages, mainly cytotoxicity [50]. At the same time, biological crosslinkers may have few drawbacks, for example, genipin induces a dark color [50]. In order to avoid the cytotoxicity caused by chemical crosslinking reaction, polysaccharide and protein multifunctional derivatives were tested as potential crosslinkers, exhibiting good cellular compatibility and proliferation [46].

Strong physical networks can be induced by helicoidal structures, the presence of glassy nodules or crystalline domains. Weak physical networks are formed by hydrogen bonds, molecular specific binging, metal–ligand coordination, ionic, hydrophobic, host–guest or antigen–antibody interactions, π-π stacking, chain entanglements, thermal or pH induced gelation, and protein or poly(peptide) interactions.

Of high interest are the structures with reversible covalent bonds, such as imide, disulfide, acylhydrazone, and borate ester bonds. Under an external stimulus, these dynamic bonds are cleaved, and they can be reestablished when the action of stimulus is removed; such smart materials present intrinsic self-healing ability. The recent studies established new aspects for a deep understanding of soft and wet matter, including molecular transportation, mechano-chemical reactions, nonlinear response to large deformation, stress relaxation, and non-equilibrium processes [56].

The functionality of soft living tissues is due to specific structure and high water content. The induction of improved performances and new functionalities to hydrogels realized by analogy with biological materials is a continuous challenge. Over the last years, considerable progress was registered in the bioinspired hydrogel approaches. Biosystems are able to produce structures with complex functions, well-organized at different length scales. Researchers are looking in nature for inspiration for developing functional hydrogels with controlled structure and tuned properties in order to satisfy the increasing demands of practical applications. To achieve these requirements, a combination of self-assembly, chemical crosslinking, functionalization, or phase transition processes were used to produce artificial tissues or soft artificial organs for substituting damaged artificial organs (especially blood vessels or artificial heart) or to improve the functions of living tissues [44,46,56,58,59,60,61].

*The self-assembling* procedure represents a high versatile approach inspired by nature, used to modulate the network structure, in order to obtain the desired biological, chemical, or mechanical characteristics. The supramolecular self-assembly offers the possibility to obtain a variety of biomaterials for improving everyday life with a minimum impact to environment. Thus, different polymeric architectures and morphologies were designed as drug carriers for controlled release and optimized targeting [6,41,58,59,62,63], wound healing materials [24,39,40], tissue engineering [4,5,36,37,38,46,52,57], healthy food [19], or eco-friendly packaging [48,61]. In addition to the functional performances, the morphological stability and structural integrity of biomaterials determine their in vivo effectiveness [24,25,26,27,28,29,46,53,54].

In *semi-IPNs* or *full-IPNs*, a natural or synthetic entangled polymer is entrapped in another matrix, without covalent bonds between the two polymers. These structures present a synergistic combination of physico-chemical characteristics of the two polymers, presenting tunable physical properties, improved mechanical strength, enhanced drug loading and delivery, and so on. IPN is obtained by adding a monomer in aqueous solution of the first polymer, and then the monomer is polymerized [29,64], whereas a semi-IPN is formed by physical interactions (through –CONH_2_, –OH, –COOH, –NH_2_ groups) between macromolecules (natural or synthetic polymer, protein) and a crosslinked network [65].

By selective crosslinking of chitosan with epichlorohydrin in alkaline medium and simultaneously partially hydrolyzing an amide group of polyacrylamide (PAAm) matrix (to generate anionic sites), a full-IPN was obtained [66]. Chitosan and poly(vinyl alcohol) form full-IPNs with mechanical strength and resistance to wear [67]. Poly(methacrylic acid)/gelatin hydrogels, with tunable structural, physicochemical, mechanical, and biological properties were obtained by free-radical polymerization of methacrylic acid in presence of gelatin [68]. These hydrogels were pH-sensitive with compressive strength up to 16 MPa and allowed the cell proliferation.

*Double-networks* (DN) are elaborated for enhancing the mechanical (high strength and toughness) and other properties of hydrogels in order to meet the growing demands that a material must have in various fields [69,70,71,72]. Usually, the first network has sacrificial bonds and provides mechanical strength and rigidity, during high deformation dissipating a large amount of energy; the second network is soft and ductile (weakly or non-crosslinked) and makes the hydrogels flexible and tough, it fills inside the first network, absorbing the external stress [73,74]. A few DN hydrogels were reported to have high bulk and interfacial toughness, for example, agar/poly(N-hydroxyethyl acrylamide) DN (≈7 kJ/m^2^) [75], superadhesive poly(2-acrylamido-2 methylpropanesulfonic acid)/PAAm (1 kJ/m^2^ and bonding strength of 1 kN/m) [76], and poly(vinyl alcohol) (PVA) based hydrogels (0.8 kJ/m^2^) [77]. Fully physically agar/poly(N-hydroxyethyl acrylamide) DN hydrogels achieve high toughness in bulk and on solid surfaces (glass, titanium, aluminum, or ceramics). The efficient energy dissipation due to the reversible hydrogen bonds in bulk of both networks allows to achieve high values of strength (fracture tensile stress of 2.6 MPa), interfacial toughness (≈7 kJ/m^2^, beyond the values of tendon or cartilage bonds) and extensibility (fracture strain of ≈8), recovering 62% of toughness and 30% of stiffness after two hours exposure at high temperatures [75,78].

A *triple-network* as a stiff and resilient hydrogel was obtained by using Bombyx mori silk fibroin, gelatin and carboxymethyl cellulose (CMC) [79]. The resilience was attributed to CMC presence and the dynamic behavior of the hydrogel was due to β-sheets formation in the silk. These injectable hydrogels are able to support mechanical load-bearing joints, enabling chondrogenic differentiation of stem cells. Simultaneous control of stiffness and contraction in time mimics cells condensation and facilitates cartilage development. Triple-cationic cryogels with a high mechanical strength and excellent sorption of phosphate were obtained with crosslinked CS, low molar mass branched poly(ethylene imine) (PEI), and either PEI or poly(N,N-dimethylaminoethyl methacrylate) networks [80].

Hydrogels incorporating *dynamic interactions* (hydrogen bonds, metal–ligand coordination, host–guest molecular recognition, etc.) exhibit special properties, such as mechanical strength, shape memory, self-healing, and easy processability. Dynamic characteristics can be tuned through either spatial (topological architecture) or temporal (through crosslinking kinetics) hierarchy. Cucurbit[*n*]uril (CB [8]) macrocycles are used as host molecules (able of accommodating simultaneously two guests: an electron-deficient and another electron-rich guest) [81]. Host–guest complexation between CB [8] and functionalized copolymers allows to obtain dynamic biocompatible hydrogels with shear-thinning behavior [82]. The temporal and structural hierarchy is influenced by the dynamics of the transient crosslinks: a fast relaxation mode with a lifetime corresponding to the crosslinking kinetics of nonentangled macromolecules and a slow relaxation mode for the entangled state. Water soluble polyrotaxanes were obtained by using poly(ethylene glycol)-amine, CB [7] and 2,3,6-tri-O-methyl α-cyclodextrin [83]. These inclusion complexes are able to encapsulate neutral and ionic guests. A dynamic hydrogel structure was obtained from highly branched CB [8]-threaded polyrotaxane (spatial structure influencing the macromolecular relaxation) and linear naphthyl-functionalized hydroxyethyl cellulose, with improved mechanical and viscoelastic properties. The spatiotemporal hierarchy allows engineering the dynamics of networks with superior performances, such as higher viscoelastic moduli, improved thermal stability, and self-repairing/healing ability [84].

Transition metal ions have the ability to catalyze the enzymatic processes specific to the biological organisms. For example, marine mussels or worms secrete in aqueous environment mechanically robust materials with hierarchical structure, with high toughness (high stiffness and high extensibility) and with self-healing ability. Such a structure is formed by *dynamic metal-coordination bonds.* By mimicking the capacity to break and reform of natural metal-coordinate complexes, materials with tunable and reversible mechanical properties were created, from tough gels to soft robots, by introducing either permanent or sacrificial bonds involving Fe^3+^, Zn^2+^, or Ni^2+^ [85]. The kinetic lability of these bonds, higher than that of the covalent bonds, determines the occurrence of dynamic and reversible mechanical properties, such as enhanced toughness, due to the continuous viscous flow and network remodeling. A rigorous control of the relaxation times of materials (through the metal aggregation in the network) will improve the cell growth and differentiation, drug delivery, or therapeutic efficiency.

During a long period of time, the composites with hierarchical structures evolved at micro- and macroscale to improve the mechanical performances and functionality in order to ensure their resistance in their daily environment found in the living organisms. Toward synergism of the multicomponent systems, the anisotropy and predictable response to external stimuli is also important in the hydrogel design. Thus, ligament-inspired bilayer fibrous gel belts with high mechanical strength were designed via dynamic stretching. The anisotropic fibrous gel belts with programed shape deformations were obtained by polymerization of monomer/nanoparticle precursors in glycerol/water (1:4) solution and dynamic stretching on the formed pregels (2-(2-methoxyethoxy) ethyl methacrylate, and oligo(ethylene glycol)methacrylate and isopropylacrylamide were used as monomers) [86] as promising materials for smart actuators, biomimetic devices, or soft robots.

Beside their tunablephysiochemical properties and biocompatibility, hydrogels present the advantage of *very low friction* [87]. Three regimes of frictional behavior were identified for poly(acrylic acid) (PAA), polyacrylamide (PAAm), and agarose hydrogel spheres on smooth surfaces. At low velocity, it was found that the friction is controlled by hydrodynamic flow through the porous hydrogel network, being inversely proportional to the characteristic pore size. At high velocity, the lubricating film formed between the gel and the solid surface obeys the elastohydrodynamic theory. Between these two regimes, the friction forces decrease by a decade and display slow relaxation over a few minutes. This was attributed to interfacial shear thinning behavior of polymers and to increase in the relaxation times due to the confinement of entanglements. The transition between different friction regimes was tuned by salt addition, solvent viscosity, or sliding geometry at the interface.

Inspired by natural articular cartilages with excellent mechanical properties and surface lubrication, composite materials were developed by grafting hydrophilic polyelectrolyte brush layers on the surface of hydrogels [88]. A friction reduction was also obtained by introducing Tween 80 into PVA physical or chemical hydrogels [89]. Benzophenone was entrapped on the surface of polydimethylsiloxane, polyvinylchloride, or polyurethane and acrylamide was polymerized by ultraviolet irradiation; a hybrid surface with low friction under very high load conditions was obtained, able to conserve the lubrication and wear performances over 100,000 cycles under a pressure of about 10 MPa [90]. Muscle-like properties with aligned nanofibrillar architectures were obtained in the case of 3D-printed PVA hydrogels prepared by freezing/thawing method: high fatigue threshold (1.25 kJ/m^2^) and strength (1 MPa), values of Young’s modulus around 0.2 MPa and about 84 wt.% water content [91].

### 2.2. Tunable Characteristics of Hydrogels

The main aspects in selecting the appropriate material for a specific application refer to morphological and structural aspects that influence the physico-chemical and biological properties of hydrogels. Hydrogels are suitable for biomedical applications due to their high swelling in water, porous and permeable structure able to incorporate drugs, proteins, living cells, or bioactive factors. The characteristics of physical or chemical hydrogels are mainly influenced by the nature of the network and its constituents and by a crosslinking degree that can be controlled during synthesis. The swelling and mechanical behaviors, as well as the biological properties, are analyzed as a function of physico-chemical structure of the network, ionic charges, the nature of crosslinkers and degree of crosslinking. Several of these aspects will be briefly pointed out in the next sections.

#### 2.2.1. Porosity

The most important characteristics of the hydrogels are the average pore size, the distribution of pore size, pore type (open or closed) and their interconnections, depending on crosslinking density (influenced by the concentration of crosslinking agent, physical entanglements or net charge of polyelectrolyte) and composition (content and type of macromolecules, solvent). Due to their porous structure and swelling properties, hydrogels present a high elasticity, close to natural tissue, ensuring the transport of nutrients, proteins, peptides, oligonucleotides, minerals, and drugs, as well as cell proliferation. Usually, the porosity is estimated by electronic microscopy (SEM, TEM) [40,92].

As an example, the SEM images of PVA/HPC/BSA (HPC-hydroxypropyl cellulose, BSA-bovine serum albumin) physical hydrogels (Figure 1) revealed a porous network structure consisting of interconnected pores with the average size between 19.3 μm and 26.7 μm, influenced by the composition and the number of applied freezing/thawing cycles [40].

#### 2.2.2. Swelling

Hydrogels are able to absorb water up to several thousand times their weight in dry state, without dissolving. The first water molecules come in contact with the network hydrate, the hydrophilic groups, and create the primary bound water, and the hydrogel starts to swell. Then, the secondary bound water is created by the hydrophobic groups. The gel will incorporate an additional amount of water due to the osmotic driving force (the pores created in the hydrogel bulk enhance the absorption of water by capillary forces) until the swelling equilibrium is reached, when the pores or voids are filled with water. If the crosslinking points are established by weak physical interactions, the water excess can determine the network disintegration up to complete dissolution. The amount of free and bound water can be determined by using different techniques and methods, such as: nuclear magnetic resonance, differential scanning calorimetry, X-ray powder diffraction, dielectric relaxation spectroscopy, quasi-elastic neutron scattering, infrared spectroscopy [93,94], diffusion or adsorption [40,41,59,62,63,95,96]. The main factors influencing the swelling process are: the crosslinking density, hydrophilicity of polymer chains, and characteristics of surrounding medium, such as pH or ionic strength and temperature. The dynamics of swelling and diffusional solute release from hydrogels are well-discussed by Peppas et al. [59,62,63]. The swelling/shrinking of polymeric networks in different environmental conditions is the key behavior of smart systems able to abruptly change their volume under the action of physical, chemical, or biological stimuli (Figure 2).

#### 2.2.3. Biological Properties

The main requirements for a hydrogel as biomaterial are: biocompatibility, nontoxicity (biologically safe), and ability to be sterilized [97], among other properties that are presented below. Some demands must be met simultaneously, so a suitable material must have multiple functions [3]. The natural biomolecules (polysaccharides, proteins, and essential oils) endow usually good biological performances to hydrogels, including antioxidant, antibacterial, antiviral, anti-inflammatory, and antifungal properties. These characteristics are mainly included in biomaterials as (1) inherent biological properties (for networks formed only by biomolecules); (2) biological properties of composite hydrogels that incorporate natural and synthetic compounds or networks with chemically modified polysaccharides or peptides; and (3) systems containing only synthetic (macro)molecules able to form biocompatible and non-toxic network with biological properties [98,99,100].

*Biocompatibility* is a critical factor for any biomaterial used in contact with a living tissue, the property to respond properly in physiological environment, to be bio-safe (non-toxic), and to present biofunctionality [55]. There are two types of biocompatibilities: bulk (mechanical biocompatibility and design biocompatibility) and interfacial (typical interfacial biocompatibility, blood compatibility, or tissue connectivity) [100]. Good mechanical biocompatibility means that the material has the same mechanical properties as the organs. When the biomaterial surface comes in contact with blood, there are some complex biological changes that limit the use of many materials. Heparin is a very effective anticoagulant (it prevents blood clots at the material surface), but its use is reduced in the case of long exposure times because possible side effects (internal bleeding). Hydrophilic and hydrophobic groups are grafted to the hydrogel surface and ionic interactions ensure the protein interactions. Adhesion of connective tissues is considerably improved by using a thin collagen hydrogel film [101,102].

Biocompatible hydrogels result by self-assembling dipeptides (the shortest unit in peptide self-assembling) as a result of coordinated noncovalent interactions by considering molecular (geometry, hydrophobicity, electronics) and environmental (temperature, solvent, light, external forces, etc.) factors [103]. The main chain of dipeptide hydrogelator is obtained by self-assembling via amide bonds and the side chains associate through π-stacking interactions or hydrogen bonds in into viscoelastic fluid or 3D interfacial network under appropriate conditions [104]. The self-assembly approach can be applied to various polysaccharides (chitosan, alginate, hyaluronic acid, carrageenans, agarose, etc.) or proteins (collagen, gelatin, fibrin, elastin) for tuning their amphiphilic behavior via host–guest interactions in order to develop soft materials with anisotropic structures that enable cells or oils encapsulation with potential applications in cosmetics, tissue engineering, therapeutics, food industry, etc. [46,81,105,106].

*Antioxidants*. Oxidative stress produces significant toxicity influencing the cellular transplant. Antioxidants are encapsulated inside the hydrogels to inhibit free radicals. Usually, they are natural (poly)phenols, able to undergo oxidation/reduction reactions [107,108]. The effective inhibition of bacterial infections is also an important characteristic of biomaterials.

*Antimicrobial agents*. Hydrogels loaded with antibiotics, metal nanoparticles (NPs), antimicrobial polymers, and peptides can release the antimicrobial agents in a sustained manner, which is important to treat infections effectively and prevent biofilm formation. Biodegradable antimicrobial polymer-loaded or peptide-loaded gels are more attractive than gels encapsulated with antibiotics or NPs because antibiotics easily develop drug resistance, and it is relatively more difficult to mitigate toxicity of metal NPs due to their non-degradability.

The hydrogels could present either intrinsic antimicrobial properties or may act as carriers for antibiotics. Antimicrobial molecules kill the pathogenic microbes inside the cells. Thus, the design of hydrogels for antibacterial materials represents a high interest for many investigators, as can be seen in recent publications [109,110,111,112,113,114]. The antibiotics used for antimicrobial hydrogels are discussed by Yang et al. [110]. Combined with other antimicrobial materials, the hydrogels with antibiotics revealed improved antimicrobial properties and biocompatibility.

Antimicrobial peptides defend plants and animals and present a broad antimicrobial spectrum, with small or no bacterial resistance. To limit the enzymatic degradation, some strategies were considered: the incorporation of d-amino acids, modification of peptide structure either to side chain groups or N- and C-terminal, cyclization [115]. Some studies take into account the conjugation of the antimicrobial peptides with proteins, synthetic (Pluronics), or natural (polysaccharides) polymers in order to improve the therapeutic efficiency of antimicrobial peptides [116].

Bioactive antibacterial peptides, quaternary ammonium compounds, metal-oxide (metals: silver, gold, zinc, cooper) NPs, CS, gelatin, or antibiotics (ciprofloxacin, gentamicin, vancomycin), biological extracts (propolis honey, lavender, peppermint, thyme, rosemary, cinnamon, eucalyptus, pomegranate seed, lemongrass) are used to induce resistance against different bacteria [98,99,117]. Using the Schiff base reaction, a thermo-responsive hydrogel of Pluronic F127, poly(ε-lysine) and hyaluronic acid was used for loading an antibacterial photocatalyst, used for continuous sterilization and bacteria-infected wound healing [118]. Many efforts are focused on alternatives to antibiotics and bacterial resistant materials [119].

Inflammation is the reaction of the body as response to the harmful stimuli, such as pathogens, irritants, or damaged cells. There are some anti-inflammatory drugs that reduce the inflammation or swelling, but in excess they produce side effects (gastrointestinal or kidney problems). In order to minimize these effects, hydrogels based on peptides, collagen, chitosan, or hydroxyethyl cellulose, are used as localized drug depots [98,120].

*Antiviral* effects were gained by incorporation of antibiotics or antiviral drugs in polymer hydrogels, as well as by mixing polymer networks with antimicrobial compounds, such as peptides, lipids, surfactants, and NPs [99]. Biocompatible, pH sensitive IPNs of CS, xanthan, and poly(2-acrylamido-2-methylpropane sulfonic acid) were tested for antiviral drug release (acyclovir) [121]. For the same drug, it was proposed an IPN of acrylamide-grafted dextran and chitosan [122]. The hydrogels obtained by freezing/thawing of PVA/Laponit, loaded with Rifampicin, were recently investigated as a promising drug delivery system for COVID-19 treatment [123].

A broad spectrum of *antifungal activity* is given by including some antifungal agents into hydrogels, such as Amphotericin B (against *C. Albicans*), terbinafine hydrochloride (against *Candida Krusei* and *Candida Albicans*), or peptide, resulting formulations with wound healing properties [98,124,125]. Thus, hydrogels of PEG with citric acid and indole-3-acetic acid present antifungal activity against *Aspergillus Fumigates*, *Rhizopus Oryzae*, and *C. Albicans* [3,88]. Biocompatible dextran-based hydrogels containing Amphotericin B kill efficiently *C. Albicans* within two hours, with effect for minimum 53 days [125]. Such hydrogels are suitable materials as catheter coatings in order to avoid intravascular catheter infections. By incorporation of terbinafine hydrochloride (cationic drug) into anionic composite hydrogels, such as poly(acrylamide/maleic acid) [126], poly(N-vinyl 2-pyrrolidone/itaconic acid) [127], polycaprolactone/gelatin (50:50 wt./wt.) [128], and pH-triggered antifungal action was observed.

#### 2.2.4. Mechanical Properties

Mechanical properties are very often investigated for hydrogels, being connected with the durability and particular requirements in the conditions in which the material may be used. The response of material to external stress is very important for 3D printing materials, but also for injectable hydrogels, tissue engineering, cartilage replacement, tendon or ligament repair, wound dressing, drug delivery matrix, etc.

Generally, the mechanical characteristics of materials are studied in terms of stress–strain (or load–deformation) relationship using specific techniques, such as rheological measurements, compression and tension tests or dynamic mechanical analysis. Submitted to the action of external forces, polymeric networks exhibit viscoelastic feature, i.e., both viscous and elastic characteristics during deformation. The behavior of the hydrogels at different time scales are investigated through the viscoelastic moduli as a function of time, oscillation frequency or strain: G′ is the elastic modulus, as a measure of the stored energy quantifying the network rigidity; G″ represents the loss modulus, as a measure of the dissipated energy during a shearing cycle, characterizing the viscous flow; G′ >> G″ for solid-like behavior (typical for hydrogels) and G″ > G′ for liquid-like behavior. The ratio G″/G′ is known as the loss tangent (tanδ) and it expresses the degree of viscoelasticity for a sample [129]. The performances and potential applications of hydrogels can be discussed considering their stiffness (correlated with G′ magnitude) and tan*δ* values [130]. Samples with liquid-like behavior are predominantly viscous, G″ > G′ and tan*δ* > 1. Highly entangled polymers and self-assembling systems present tanδ close to 1. Near the gelation point, G′ ≅ G″, tan*δ* ≅ 1 and the so-called critical gel state is reached [131]. The value of the loss tangent becomes slightly below 1 for networks with very low crosslinking degree, it continues to decrease as the hydrogel structure is formed and the solid-like behavior becomes preponderant (Figure 3) [132]. The loss tangent may be a tailorable property for hydrogels obtained by physical or chemical crosslinking, providing a quantification of viscoelastic response of network to loading. The most hydrogels reported in the literature present tan*δ* value around 0.1 and very stiff networks. For example, the hydrogels composed by gelatin methacrylamide, gellan gum, and methyl cellulose [133] have tan*δ* values between 0.01 and 0.1, as in the native tissue.

Creep and recovery curves give a clear differentiation of the viscoelasticity in solution (when tan*δ* >> 1) and hydrogel state (tan*δ* ≅ 0.1), as observed in Figure 4. Low shear stress values applied to a PVA/HPC aqueous solution determine a high permanent deformation (γ > 100), the solution behaves as a pure viscous fluid and there is no recovery of deformation when the external forces are removed (Figure 4a). In similar conditions, the physical hydrogel obtained by freezing/thawing of the same PVA/HPC solution exhibits very small deformation during creep (γ < 0.01) and its structure is completely recovered in less than 300 s (Figure 4b), displaying instantaneous and delayed elastic deformation. For higher shear stress (as for example 50 Pa), the viscoelastic response of the hydrogel contains instantaneous and delayed elastic deformation, but also includes the permanently viscous (irreversible) part.

Gelation kinetics, rigidity, thixotropy, and structure recovery are usually investigated for establishing the hydrogel performances required in the targeted applications [27,35,40,41,92,129,130,131,132,133,134]. Due to their composition, structure at different levels and functions, there are many types of hydrogels determining a large variety of viscoelastic responses. The swollen strong hydrogels exhibit elastic behavior, i.e., high extensibility when they are submitted to deformation and complete recovery after cessation of deformation. However, most of networks present time-dependent viscoelastic behaviors. Thus, the approach considered for a hydrogel design, depending on the targeted application, considers the relationship between the viscoelastic properties and physico-chemical structure of the network (hydrogel composition, crosslinking density, mesh size of the network, the frequency and strength of different interactions). Different rheological tests, such as temperature [134], time [24,134], deformation or frequency sweep [24,27,35,40,135], creep and recovery [27,28,29,134,136], continuous shear measurements [27,40] are often required. The usual rheological tests, in correlation with the hydrogel structure, are presented in a recent review [137].

The structure in a particular macromolecular network in static state, created by a certain preparation protocol, is usually correlated to rheological parameters at very low deformation. This may be different from the structure adopted by the network submitted to large deformation (shear or extension) when self-organization or degradation can be induced by the applied forces. The occurrence of non-linear effects at different levels has to be considered: weak effects due to weak interactions that determines changes in the relaxation spectra of viscoelastic fluids; strong effects when a new equilibrium state is reached due to structural changes; flow instabilities or phase transition changes due to thermodynamic considerations [138]. In addition, the time dependent phenomena associated to linear viscoelasticity are correlated to material structure [137,138,139,140].

Many efforts are dedicated to the formulation of hydrogels with structural and mechanical homogeneity and long-term stability. The mechanical properties can be improved by physical/chemical crosslinking, IPN formation, or by forming metal-coordinate complexes. One-pot synthesis of double network was reported by combining polycondensation of tetraethoxysilane and radical polymerization of dimethyl acrylamide, resulting in a lubricant gel stable for long time periods at high temperatures [141]. For this gel, a higher compressive fracture stress value (>4 MPa) was obtained, as compared with that of the gel resulted from the two-step synthesis (3 MPa). For tissue engineering applications, soft materials which resemble the biological extracellular matrix (ECM) are required. These materials may ensure tissue-like mechanical properties and an efficient nutrient and waste transport. ECM serves as mechanical support required for cell growth into tissues, providing adhesion sites, tissue development, and wound healing. Cell adhesive, biocompatible polyurethane-based hydrogels with high swelling ratios and distinctive thermal sensitivity represent versatile materials with tunable properties for tissue engineering and other applications [41,53,141,142].

PVA-based hydrogels provide also promising platforms with a broad range of tunable mechanical properties [24,27,40,41,49,92,96,129,132,134,142,143,144,145]. By choosing the proper method of preparation and the optimum composition, the mechanical properties of hydrogels can be modulated over a large window: tensile strength from 10 kPa to 10 MPa, toughness from 100 J/m^2^ to 100 MJ/m^2^, elongation in the range 100–2100%, and elastic modulus from 1 kPa to 2.5 MPa.

Lu et al. [146] obtained silica-fiber-reinforced PAAm/sodium alginate (Alg-Na) hydrogels with multiple sensing characteristics (stretching, compressing, and bending). Interfacial chemical bonds were created in situ between silica nanofibers and chains formed by free-radical polymerization. The reported hydrogels present enhanced ionic conductivity (3.93 S/m) and improved mechanical properties in terms of tensile stress, strain, modulus, toughness, being promising materials for tissue engineering, medical devices, wearable electronics, bioactuators, etc.

Soft, stretchable, and elastic multifunctional hydrogels present a high interest for flexible wearable biosensors triggered by mechanical forces or pressure. An ionic conductive organogel was prepared by copolymerization of acrylic acid and cardanol in water/1,3-butanediol mixture. The addition of 1% cardanol in the monomer system increases several times the tensile strength and toughness and improves the resilience of PAA. This system can act as a sensor to detect human motion in real time (wrist, finger, elbow, and knee joints) [147]. Inspired by biological tissues, wearable biosensors with mechanical properties similar to the skin (high elasticity and strength, antifatigue, and self-healing ability) were fabricated using SiO_2_-based particles as crosslinking points for the networks. Thus, SiO_2_-*g*-poly(butyl acrylate) (SiO_2-_*g*-PBA) core–shell particles were used as hydrophobic dynamic crosslinkers for poly(acrylamide-*co*-lauryl methacrylate) (P(AAm-*co*-LMA) [148,149] or DN formed by PAAm, as a first network, and alginate crosslinked in presence of Ca^2+^, as second network [150]. It was shown that the dynamic networks dissipate a high amount of energy during destruction and recombination, achieving a wide spectrum of mechanical properties, exceptional fatigue resistance, and good conductivity in the presence of salts. These hydrogels are suitable as biosensors for health monitoring, human−machine interaction, or voice recognition. Thus, these hybrid hydrogels were integrated into different parts of the human body and evaluated as potential wearable or pressure biosensors and for phonation recognition (Figure 5). The relative resistance changes (Δ) were calculated as 100 × (R − R_o_)/R_o_, where R_o_ is the resistance in absence of strain and R is the real-time resistance under the stretch [149].

Other nanocomposite hydrogels, with high stretchability (1600%), rapid recoverability, adhesiveness, self-healing ability (96.5%), and gauge factor of 5.86, were obtained by polymerization of acrylic acid in the presence of a dynamic crosslinking center provided by core–shell hybrid NPs SiO_2_-*g*-PAAm [151]. The gauge factor is the ratio between the relative resistance (ΔR/R_o_) and the applied strain (γ). The hydrogen bonds between PAA and PAAm chains can reversibly be destroyed and reformed, dissipating a high amount of energy. The characteristics of these hydrogels make them suitable as sensors for human motion monitoring or artificial skin.

Micelle-like aggregation by self-assembly of alkyl-modified PEG in aqueous solutions provides dynamic junctions for hydrogel formation during polymerization of acrylamide in presence of methylene bisacrylamide. The reversible break/reformation of micelle-like aggregates upon stretching and the presence of high entangled system containing PEG and poly(acrylamide), and hydrophobic domains involving the alkyl terminal groups of PEG generate a dynamic network with super-stretchable properties and high toughness, suitable for flexible sensors, artificial skins, and drug delivery systems [152]. Wearable sensors were recently reported using PAA with cellulose whiskers, in presence of tannic acid and Fe^3+^ in glycerol/water mixture [153] or PAA with lignin in CaCl_2_ solution [154] for health detection, electronic skin, or human–machine interface.

Thermal-sensitive bilayer actuator was prepared using poly(N-isopropyl acrylamide)/cellulose nanofiber (PNiPAm/CNF) DN hydrogel as the top layer, and poly(acrylamide-*co*-acrylic acid)/cellulose nanofiber (P(AAm-*co*-AA)/CNF) double network hydrogel as the bottom layer [155]. By rapid variation of the temperature, this actuator quickly changes the weight (about 70 times), the tensile strength reaches ≈ 730 KPa and elongation at break reaches ≈ 410%. Further crosslinks in presence of Fe^3+^ increases the tensile strength to 3.86 MPa. Another system composed by P(AAm-*co*-AA) and oxidized sodium alginate forms smart ionic hydrogels in the presence of FeCl_3_, showing an accurate detection of human movement (gauge factor of 7.8) [156].

*Stiffness* is an important structural clue for hydrogels used as tissue engineered substrate, correlated with scaffold dimensionality and cell types [157], affecting the cell adhesion, migration, differentiation, and ECM morphology. ECM is a dynamic network with molecular composition and matrix pliability modulated by the resident cells. Inside the human body, the ECM stiffness (measured through Young’s modulus, a measure of the resistance to deformation) presents different values (see Figure 2 from [157]). Thus, it varies from brain (1–3 kPa) and muscle (23–42 kPa) to tendon (136–820 MPa) and bone (15–40 GPa). The bone is the hardest human tissue (15–40 GPa) [157,158], whereas for the demineralized bone matrix the stiffness decreases (~0.67 MPa [157,159]); the aggregation of endothelial cells in blood vessel is superior in stiff matrices (37.7 kPa) compared to soft matrices (13 kPa) [160]. The strategies adopted for regulation of the substrate stiffness were recently discussed [161]: controlled crosslinking density, molecular weight, and concentration of different constituents in multicomponent hydrogels; tunable intermolecular interactions, incorporation of NPs, design of the appropriate architecture using an adequate approach. Hydrogels with tunable stiffness were fabricated from different materials and the effect of their stiffness on tissue repair was analyzed [157,161].

Similar to the dermis of the sea cucumber, the variable stiffness is important for living organisms. Inspired from the dermis, stiffness-changing smart materials with interchangeable hydrogels with switchable mechanical properties were developed [162]. The double-stranded supramolecular network showed reconfiguration-dependent self-healing behavior and tuned formability, being a promising biomimetic material with high durability.

#### 2.2.5. Self-Healing Ability

Self-healing is a remarkable function of biological systems consisting in self-repairing of diverse types of damages. Learning from the biological systems, the great challenge for researchers is the development of bioinspired self-healing materials, able to spontaneously repair themselves after damage or degradation and to recover structural integrity and functionality [163,164].

The mechanism of healing for the polymers and their composites is based on intrinsic or extrinsic methods. The intrinsic healing is an inherent capability of the material and it supposes the existence of chemical, physical, or supramolecular interaction between the polymer chains which can break and then easily reform. The extrinsic approaches use a healing agent that is stored in reservoirs (microcapsules, hollow fibers, vascular systems, etc.) and ensures the repair of the damaged structure.

Many efforts were carried out to design and to characterize the performances of polymers and materials by triggering and tuning self-healing behavior, analysis of recovery properties and assessment of healing efficiency [10,12,14,24,25,26,27,28,29,103,114,146,147,151,152,157,165]. The hydrogels can be used as biomimetic systems for understanding the complex cells behavior to various microenvironments. The major challenge was to reproduce the 3D architecture and dynamic mechanical behavior of native extracellular matrices. A hydrogel of four-armed PEG-maleimide crosslinked by Dronpa145N (fluorescent protein) was developed as an efficient system to control the behaviors of cultured cells. The mechanical properties are fast and reversible changed by a photo-induced switch between the tetrameric and monomeric states [166].

Hydrogels with dynamic character and superior mechanical properties were prepared by using telechelic multiblock copolymers of hydrophilic PEG by incorporating self-complementary hydrophobic ureidopyrimidinone (UPy) moieties [167]. These networks exhibited strong pH- and temperature-responsive gelation due to strong self-associative tendency of UPy into dimers by four-fold hydrogen bonding. Due to catechol-iron complexes, the outer cuticle of mussels presents extraordinary characteristics, being adhesive, hard, stiff, and extensible [168]. Inspired by the cuticles of mussel byssus, a dynamic network, able to induce microphases with different physical crosslinking densities (high and low number of hydrogen bonds), was designed [168,169]. Reversible sacrificial bonds were incorporated in soft domains (Figure 6) in order to achieve high toughness (up to 62 MJ/m^2^) and self-healing properties (recovery against 650% elongation), whereas the hard domains avoid the material deformation (tensile stress about 17 MPa).

Polysaccharide and protein-based hydrogels are promising therapeutic materials due to their biocompatibility and biofunctionality. In addition, the dynamic crosslinked networks represent very attractive hydrogel platforms due to easy and eco-friendly fabrication, versatile structure, and self-healing ability [24,25,26,27,135,136,165].

Self-healing properties of the hydrogels can be analyzed through their thixotropic behavior in different shear conditions. As for example, Figure 7 shows how the viscoelastic parameters are changed for a protein/polymer hybrid hydrogel when it is submitted to several cycles of low (1%) and high (100%) step strains. The hydrogel contains 50% protein (BSA), 50% polymer (mixture of 10% HPC and 90% PVA), and 1 mmol/L reduced glutathione [24].

The rest structure is preserved when a low strain (γ) of 1% was applied, G′ >> G″ indicating a solid-like behavior. Then, by sudden increase in the strain to 100%, G′ and G″ moduli decrease rapidly due to the temporary disruption of physical interactions and G′ < G″, showing a liquid-like behavior. By decreasing the strain to the initial value of 1%, G′ and G″ return to their initial values. After running three cycles of 1–100% step strains, no significant weakening in network strength was observed. Such hybrid hydrogels preserve their structure for a high number of cycles at increasing deformations (Figure 8) without failure of their network structure or self-healing ability.

Various methods and equations were tested for evaluating the quality of self-repair. Generally, the mechanical tests (such as: cycling deformation, tension, compression, torsion) are used to determine the material properties (stiffness, strength) before and after self-repair. In most cases, a dimensionless value of healing efficiency (HE) is calculated, expressing the recovery percentage of the analyzed mechanical property as the ratio (or as percent) between the healed value to pristine value of property or (healed–damaged)/(pristine–damaged) values of property. The development of reliable validation methods for HE is still a subject of debate [170]. Monitoring both self-healing ability and structural functionality of the overall structure may give information on the material quality or its state of being functional.

Efforts were undertaken to mimic the enzyme-regulated self-healing from biological systems by the development of a suitable platform for the manufacture of enzyme-assisted polymeric hydrogels. The enzymatic crosslinking strategy is an innovative alternative to the other crosslinking procedures. By controlling the dynamics of reversible physical interactions or chemical bonds, enzymatic reactions can be directed towards the healing of damages inside of polymeric networks [171]. Thus, enzyme-enabled crosslinking is a high selective procedure for specific enzymes and it can be performed under mild physiological conditions [172]. The most used enzymes are transglutaminase and horse radish peroxidase (HRP). Transglutaminase catalyzes trans-amidation reaction, inducing the gelation of a protein solution [173]. Horse radish peroxidase determines the polymer gelation by oxidative coupling of hydroxyphenylpropionic acid moieties [174]. Injectable hydrogels based on the glycopolypeptide were obtained through enzyme-mediated crosslinking approach in the presence of HRP and hydrogen peroxide (H_2_O_2_). Tunable mechanical properties, swelling, degradation, and gelation times were achieved by varying the concentration of HRP and H_2_O_2_ [175].

Engineering the self-healing ability, various smart composite materials can be designed for many applications, such as: healthcare (hydrogels for tissue engineering and regenerative medicine, smart drug-delivery systems, etc.), sensors and actuators, coatings, biomimetics, electronics, self-cleaning textiles, bio-inspired robotics, dental composites, polymer-modified concrete, resins and fiber-reinforced resins, concrete and cementitious composites, materials for automotive industry and aerospace engineering, etc. [176].

#### 2.2.6. Stimuli-Responsiveness

Smart hydrogels are able to undergo sol–gel or gel–sol transition (due to structural or volume phase transitions) upon subtle changes in their surroundings as response to physical, chemical, or biological stimuli [4,177]. Since their design, the properties of hydrogels can be tuned to mimic the mechanical, biochemical, and functional characteristics of the soft tissues. The biomimetic materials are applied for controlled drug delivery [4,6,41,58,59,108,122,126,127,178,179], in regenerative medicine (tissue engineering, modulating tissue environment to promote the tissue repair) [3,4,176,180], biosensors and actuators [4,9,11,12,13,86,176,180,181,182], bioprinting [4,10,11,12], 3D cell culture [4,42,46,79,177], imaging for medical diagnostics and therapy [183,184,185], personal healthcare and hygienic products [16,165], etc. A variety of hydrogels responsive to different external stimuli were reported: temperature [4,9,22,35,53,78,134,177], pH [4,65,96,121,124,132,177], electric field [4,166,186], magnetic field [4,177], light [4,177], pressure/strain [4,47,53,151], ultrasound [4], ionic strength/redox [4,177], glucose, enzyme, and antigen/antibody [4,177].

Inspired by the structural ability of chameleon skin to change the color and to regulate the body temperature under the action of external visible light, antibacterial, and photo-thermal responsive materials were developed as biosensors to detect the glucose level in the body fluids [9]. These materials are composed by two outer layers of thermosensitive hydrogels of poly(N-isopropylacrylamide-*co*-N-isopropylmethacrylamide), P(NIPAAm-*co*-NIPMAAm), that included plasmonic silver nanocubes, and an inner layer of electrospun fibers containing polycaprolactone (PCL) and polyethylene oxide (PEO), giving mechanical stability to the system. P(NIPAAm-*co*-NIPMAAm) based microgels [187] or nanofibers [188], with lower critical solution temperature (LCST) around 37 °C, can be used as a smart drug delivery platform for self-regulating the drug release. In addition, it was shown that the hydrogels with P(NIPAAm-*co*-NIPMAAm) as core and poly-l-lactide-*co*-caprolactone (PLCL) as shell mimic the native ECM and, thus, they present an improved interaction with cells.

Theranostics concept, as a combination of a predictive biomarker with a therapeutic agent, is applied for personalized cancer treatment based on a single platform able to cumulate both diagnostic and therapeutic functionalities using the molecular information from the tumors collected from patients [183,184]. Many efforts were carried out for developing nano-size or molecular-level agents serving for both diagnosis and therapy [189]. An important theranostics strategy adopted recently is the combination of therapy with the biological pretargeting approach. This can be performed by optimum combination of imaging aspects (agents or modalities) and therapeutics (therapeutic components), using components with high binding affinity and their optimal internalization by molecular complexing with the receptor, ensuring the affinity of complex for binding on the targeted cell surface [183].

## 3. Biomacromolecules Provide an Important Source for Biocompatible/Biodegradable Hydrogels

The incorporation of stimuli-responsive moieties and the increasing spatial-temporal complexity of various natural biomolecules networks allow the development of bioinspired hydrogels that need to meet more and more requirements: control of drug or biomolecule release in specific biological environment (pH, temperature, enzyme), tissue repair and regeneration, bio-inks, the detection and monitoring of hydrogels through imaging, bioelectronics, etc. Polysaccharides and their derivatives have been extensively investigated for the design of hydrogels with tunable properties. According to the recent literature data, considerable efforts were dedicated to chitosan, alginate, hyaluronic acid, pullulan, cellulose, carrageenans, xanthan, starch, dextran, pectins. The majority of the mentioned natural macromolecules present significant drawbacks including lack of structural consistency that influences the mechanical properties, difficult inter-batch reproducibility, poor stability or inadequate solubility. In the studies that will be mentioned below, a pure polymer is rarely found. Most hydrogels synergistically combine the physico-chemical characteristics of multicomponent systems, following certain dominant properties in the final formulation.

### 3.1. Chitosan and Chitin

Chitosan (CS) and chitin (CH) present in their composition d-glucosamine and N-acetyl-d-glucosamine, in form of acetylated amine groups in CH and as primary aliphatic amine groups in CS (Figure 9). They were used for many innovative biomimetic materials [29,39,42,52,66,67,96,121,122,132,190,191,192,193,194,195,196,197] due to their unique properties, depending on source of polysaccharide (external shell or cell wall, soft wings, transparent layer for eyes of insects, etc.) as well as the location and living conditions of animals/insects [198].

The high versatility of CH/CS based materials is due to complex supramolecular structure resulting from packaging of macromolecules into long fibers, process dictated by the genetic code [199]. CS can form biocompatible hydrogels, with temperature [35,200] and pH [93,132] sensitive responses, antibacterial and anti-inflammatory activities [29,165,201], and self-healing behavior [29,165].

Chitosan hydrogels have weak mechanical strength, limiting their application as biomimetic materials. To improve the mechanical properties, chemical crosslinking [93,202,203], blending with other polymers [204,205], or nanofillers reinforcement [42] were used. Duan et al., constructed a force-sensitive CS-based hydrogels by embedding the CS microgel into polyaniline and PAAm networks [206]. Cao et al. [207] fabricated a strong and tough CS-based hydrogel by double physical crosslinking, with high tensile strength of 12 MPa. Compared to natural soft tissues (cartilage or ligament), the mechanical strength in CS-based hydrogels remains weak. Using these approaches for building networks, unordered structure and moderate enhancement of mechanical properties are obtained, whereas the inherent beneficial properties of CS included into the hydrogels remain detrimental.

Bioinspired 3D constructs, as printed hydrogel scaffolds with improved mechanical performances for tissue engineering or intervertebral discs, cartilage, or meniscus, were developed by microextrusion of CS reinforced with cellulose nanofibers, with the viscosity of the printable inks between 100–500 Pa·s at shear rate of 1 s^−1^ [42]. The cellulose nanofibers present good mechanical properties and their addition into CS hydrogel gives a synergistic effect (Young modulus of 3 MPa, stress at break of 1.5 MPa, strain at break of 75%), without compromising the bioactivity and biocompatibility which is characteristic to CS.

Chitosan hydrogel with a nanofibrillar structure was obtained from alkali/urea solvent system [208,209]. Rheological measurements revealed that upon heating of entangled macromolecules, a dense irreversible network of CS fibrils is formed through association. Recently, bioinspired anisotropic CS hybrid hydrogels with aligned hierarchical fibrillar architecture, similar to those of the collagen fibers in cartilages and ligaments, were fabricated by using a pre-stretching strategy with a ductile PAAm network [205]. The hydrogel displayed excellent mechanical properties, similar to natural cartilage and ligament: tensile strength of 25.6 MPa and elastic modulus of 218 MPa along orientation direction.

### 3.2. Alginate

Sodium alginate (Alg-Na) is the salt derivative of alginic acid constituted from long linear chains of copolymers containing blocks of (1,4)-linked β-d-mannuronate (M) and α-l-guluronate residues (G) (Figure 10). This polysaccharide provides pliability, biocompatibility with hemostasis, low toxicity, and gelling ability by physical (ionic crosslinking) or chemical (in presence of chemical of or photo-crosslinking agents) methods [113,210,211].

The alginate networks hold structural similarity to the extracellular matrices in tissues, being of high interest for many biomedical and engineering applications, such as: wound healing, delivery of bioactive agents (drugs and proteins), tissue engineering, sensors, and actuators [113,210,211,212,213,214,215,216]. The M/G ratio and the number of repeating G-blocks (depending on the source alginate) are two important parameters for the gelation occurrence (only G-blocks being involved) and physicochemical properties of alginate hydrogels [113]. The gelation process can be optimized through alginate structure and gelation conditions in order to reach the elastic properties of the skin (between 1 and 10 kPa) [217].

Alginate alone presents some limitations due to poor mechanical properties, limited biodegradation or dissolution by releasing divalent ions; these inconveniences can be improved by chemical modification of alginate through oxidation [210] or by combining alginate with other natural or synthetic polymers or proteins and peptides [212,213,214,215].

Tailor-made tissue engineering scaffolds with required architecture were designed by 3D bioprinting, ensuring the incorporation of biological clues and specific microenvironments in order to support cell growth and differentiation. Bio-based nanocellulose–alginate hydrogels, suitable for 3D printing, were obtained by covalent coupling of avidin-functionalized protein to the cellulose nanofibrils and ionic crosslinking of alginate using Ca^2+^ [218]. Voids within the formed structure provided the possibility of swelling in moist and wet conditions, the hydrogels being suitable for wound healing biomaterials with internal gradients of therapeutic agents that are gradually released. Three-dimensional-printable alginate/gelatin matrix was improved by incorporating cellulose nanofibers as viscosity modifier and for enhancing the mechanical properties [219]. In addition, alginate was crosslinked in presence of Ca^2+^, improving printability and shape fidelity due to the thixotropic behavior of the samples (shear thinning and viscosity recovery). At high shear rates, a temporary destruction of the physical interactions established between the polymer chains takes place and the macromolecules are oriented along the flow direction; the rest structure is recovered by decreasing the shear rate. The addition of cellulose nanofibers increases the hydrogel strength, increasing the tensile and compressive stress as well as the value of Young modulus, but it was noticed that the elongation at break and compression yield strain decrease. High content of nanocellulose in composite hydrogels can determine the loss of structural integrity when subjected to high and continuous mechanical stress [220]. Other bio-inks nanocomposite hydrogels with alginate, tempo-oxidized cellulose nanofibrils and polydopamine NPs in composition induced significant osteogenesis [221].

The alginate layer fabricated on the surface of the poly(lactic-*co*-glycolic acid) membrane by 3D printing mimicked the dermis and was able to promote cell adhesion and proliferation in vitro, accelerating wound healing [222]. A smart calcium alginate-based fiber with pH sensitive properties was designed as a sensor for monitoring the wound healing. The fiber of hydroxypropyl trimethyl ammonium chloride CS modified calcium alginate has shown reversible color change in the pH range from 2 to 11 [213]. Skin is the first protective barrier against any external endangerment and its structure and functions must be re-established rapidly after exhibiting damages. There are several skin substitutes at different levels (epidermal, dermal, or dermo-epidermal) reported in the literature. An asymmetric construct was fabricated from CS/Alg-Na hydrogel by electrospinning and layer-by-layer 3D bioprinting [214]. According to its properties (porosity, mechanical properties, antimicrobial activity, and cytotoxic profile), this material could be efficient as skin substitute during the healing process. Another composite hydrogel dressing was developed by covalent links between carboxymethyl chitosan and oxidized alginate (aldehyde and amino groups) [215]. Gelatin microspheres embed with tetracycline hydrochloride were integrated in the network with antibacterial and biodegradable properties, enhancing the therapeutic efficiency in wound healing.

Hydrogels prepared by one-pot method using Alg-Na and PAAm presented dual physically crosslinked network formed by ionic coordination and hydrophobic association, respectively [223]. These hydrogels have shown superior mechanical properties (high values of strength, toughness, and fracture stress), self-healing ability, pH sensitivity and good thermal stability, with potential applications for biomedical applications (wearable devices, sensors, drug delivery, etc.). Dynamic DN consisting of boronic acid-functionalized laminarin and alginate were investigated for 3D constructs and tested under physiologically relevant conditions [224]. The dynamic covalent boronate ester bonds and ionic gelation in presence of divalent cations allowed a modular crosslinking with suitable rheological and mechanical properties for biomimetic 3D scaffolds. Injectable DN hydrogels were prepared from CS quaternary ammonium salt and Alg-Na by electrostatic interaction and ionic crosslinking [225]. The hydrogels presented good biocompatibility and appropriate mechanical properties for potential applications as injectable hydrogels for nerve or cardiac reparation.

### 3.3. Hyaluronic Acid

Hyaluronic acid (HA), or hyaluronan, is a linear polysaccharide composed of two alternating units, β-(1,4)-d-glucuronic acid and β-(1,3)-N-acetyl-d-glucosamine (Figure 11), with molecular weight (M) ranging from 5 × 10^3^ to 2 × 10^7^ g/mol [226].

This glycosaminoglycan is found throughout the body (about 15 g HA for a person of 70 kg [226]), in serum (low M of 10^5^ g/mol [227]), vitreous of the eye (long chains with M of 8 × 10^6^ g/mol) [227], joints, skin, and other organs and tissues of the body [226,228]. The molecular weight of HA greatly influences the biological properties: for 4 × 10^2^ g/mol < M < 4 × 10^3^ g/mol, HA activates the stress or heat shock proteins and non-apoptotic processes; for 6 × 10^3^ g/mol < M < 2 × 10^4^ g/mol, HA holds immunostimulatory, angiogenic and phlogotic activities; HA with 2 × 10^4^ g/mol < M < 2 × 10^5^ g/mol regulates biological processes such as skin repair, wound healing, embryonic development or ovulation; HA with M > 5 × 10^5^ g/mol presents anti-angiogenic activity and can play the role of space filler or natural immunologic depressant [228].

The rheological properties of HA in solution are influenced by concentration, molecular weight, and supramolecular structure. The shear thinning and elastoviscous behaviors were correlated with the destruction of hydrogen bonds and intensification of hydrophobic interactions as the shear rate rises [228,229]. The new HA hydrogels are soft and moist materials, but also it is necessary to confer a certain strength, allowing them to maintain a given shape. Due to its excellent biocompatibility and ability to form hydrogels by various chemical modifications, during the last years extensive studies were reported on HA innovative materials for various biomedical applications, such as skin biorevitalizing [230], advanced wound healing [231,232], cartilage friction reduction [233], controlled drug delivery systems [234,235], polymer scaffolds [236], and 3D bioprinting constructs [136,237].

The main disadvantage of HA hydrogels is the rapid degradation and poor mechanical stability under physiological conditions. In order to improve these characteristics, HA is used in composites with other natural [232,238] or synthetic [136,239] macromolecules. Biocomposites of HA incorporate advantageous the specific physicochemical properties of multicomponent systems and the synergic effect confers versatility, enhanced therapeutic efficacy and biological activity, as well as hemostatic performances. By co-printing HA with alginate, bioinspired 3D structures were obtained and the construct present suitable mechanical properties for articular cartilage regeneration [237]. An appropriate mechanical stiffness is required to regulate chondrogenic differentiation, thus rheological investigations are helpful for structural optimization. A storage modulus value around 8 kPa was considered optimum for the alginate-hyaluronate hybrid hydrogels to hold the cells in place, allowing their growth and proliferation [238]. These hydrogels were obtained grafting low molecular weight hyaluronate to alginate chains modified with ethylenediamine, via carbodiimide chemistry.

Crosslinking HA with polyethylene glycol diglycidyl ether (PEGDE) and tannic acid, bio-stability and mechanical properties of HA hydrogels were improved [240]. Tannic acid acts as a physical crosslinker determining the formation of strong hydrogen bonds with PEGDE. This hydrogel supports cell attachment and proliferation.

Among the biological properties, injectable HA hydrogels have the ability to match the irregular defects presenting a high potential for cartilage repair. In this respect, double crosslinked HA networks were synthesized by Diels–Alder click reaction combined with phenyl boronate ester bonds, resulting in superior mechanical properties, good injectability and adhesion, and reduced degradation [241]. Bioinspired hydrogel designed as IPN of gelatin-hydroxyphenyl propionic acid and HA-tyramine were proposed for enhancing intravitreal retinal cell therapy [242]. Horseradish peroxidase and hydrogen peroxide incorporated into the hydrogel enabled the control of gelation rate and crosslinking density.

The HA derivatives modified with crosslinkable groups and gadolinium [243] complexes undergo enzymatic degradation which can be monitored using magnetic resonance imaging. Gd- or Fe-labeled injectable HA hydrogels, as enhancing contrast agents, are suitable systems for in vivo tracking [243,244]. By means of reversible, dynamic covalent bonds, it is possible to achieve features that resembles to the dynamics ECM. From chondroitin sulfate and HA, new bio-inks were designed as DN viscoelastic hydrogels reproducing the mechanical properties of cartilage tissue [245].

### 3.4. Other Polyssacharides

*Pullulan* (PULL) is a versatile non-ionic water soluble polysaccharide consisting of α-(1,6)-repeated maltotriose units via an α-(1,4)glycosidic bonds (Figure 12), providing excellent film-forming ability, adhesive properties, and strong mechanical strength. There are some great advantages of PULL for being use in the hydrogels design: biodegradable, non-toxic, edible, non-carcinogenic, non-immunogenic, non-mutagenic, and water-soluble polymer [246]. Due to the high interest for this polysaccharide, many reviews focused on its production and applications [246,247,248,249,250,251].

The major limitation of PULL in bone tissue engineering applications is the failure to support cell adhesion and spreading, which is essential for cell proliferation and osteogenesis. UV crosslinking of PULL was performed after substituting the hydroxyl groups of with methacrylate groups [252]. PULL methacrylate was mixed with gelatin methacrylate to enhance cell adhesion, proliferation, and elongation properties. Amrita et al. [5] reported scaffolds for tissue engineering using mixtures of PULL with nanocrystalline hydroxyapatite and poly(3-hydroxybutyrate) microfibers as fillers.

The derivatization of PULL allows to introduce new functionalities and to obtain anchoring sites for different bioactive molecules of interest [27,35,49,248,253,254,255]. Porous hydrogels were obtained by combining PVA which form physical networks by freezing/thawing method and oxidized PULL which possesses the extraordinary ability to interact with –OH groups of PVA through −COOH groups [27,253]. These hydrogels preserve biological properties and do not release cytotoxic compounds. High porosity and pore interconnectivity are required in promoting cell proliferation and differentiation. The composite hydrogels have shown self-healing ability, the dynamic interactions among the functional groups help the network to preserve its structural integrity without damage after submitted to successive low and high deformation cycles.

Three-dimensional hydrogels biotin-conjugated PULL acetate NPs were designed as anticancer drug carriers and their in vivo antitumor activity was evaluated [256]. At 21 days post-injection, the tumor growth was significantly inhibited. Cefotaxime sodium loaded keratin-PULL based hydrogels were developed as dressing material in the diabetic wound [257]. The hydrogel membranes exhibit significant swelling, drug release, oxygen permeability properties, and good cytocompatibility toward 3T3-L1 fibroblast cell lines.

*Curdlan* is a water-insoluble β-1,3-glucan with a high number of –OH groups (Figure 13), able to form strong hydrogen bonds between the chains. Its derivatives (with sulfate, carboxymethyl, phosphate, or amine groups) are water-soluble and used in the biomedical field. Thus, drug delivery systems were designed by chemical [258] or physical [259,260] crosslinking curdlan derivatives. Hybrid networks of curdlan and tannic acid prepared by annealing technique showed antioxidant properties and hemostatic functions, being candidates for wound healing [261].

*Tragacanth gum* is a multi-branched heterogeneous polysaccharide containing two fractions of tragacanthic acid and arabinogalactan with good biodegradability and biocompatibility, antioxidant activity, high structural stability against environmental agents, and low cost. A bioinspired drug delivery system based on pH- and reduction-sensitive magnetic tragacanth gum hydrogel was recently developed for chemo/hyperthermia treatment of cancerous cells [262].

*Gellan gum* (GG) is a linear, anionic, extracellular polysaccharide that consists of repeating tetra-saccharide units composed of 1,3-β-d-glucose, 1,4-β-d-glucuronic acid, 1,4-β-d-glucose and 1,4-α-l-rhamnose (Figure 14), available in acetylated and deacetylated forms. It is able to undergo thermoreversible gelation and to form rigid and transparent gel, whereas in composites it gives elastic and soft gels used for cartilage substitutes, drug delivery, and intervertebral disc repair [263].

The hydrogels of deacetylated GG present improved mechanical properties upon decreasing the temperature of gellan gum solutions in the presence of Ca^2+^ and Mg^2+^, being able to support the ECM deposition and cell proliferation [264]. A biomimetic injectable thermosensitive hydrogel composed of thiolated gellan and poly(methacrylamide-*co*-methacrylate-*co*-bis(methacryloyl) cystamine) was reported by Laradji et al. [265]. This physical hydrogel is formed in-situ at body temperature, and disulfide covalent crosslinks are developed in time. The preclinical evaluation of the vitreous substitute from the tested rabbits during four months did not present toxicity and the retina functioned within the normal limits, without evidence of retinal detachments. The carboxylic acid groups of polysaccharide were reduced to hydroxyl groups, and then 2-ethyl-2-oxazoline monomer was grafted through a cationic ring-opening polymerization. The polymerization was terminated and crosslinked in situ using cystamine moiety and amine-end capped Fe_3_O_4_ as NPs. Doxorubicin hydrochloride was encapsulated into the porous hydrogel through physical interactions between the drug molecules and the polymeric network and delivered into the targeted tissue with low side effects.

κ-carrageenan (κ-carr) has a helicoidal structure of alternating (1,3) linked β-d-galactose-4-sulfate and (1,4) linked 3,6-anhydro-α-d-galactose (Figure 15), with one negative charge per each disaccharide repeating unit.

Monovalent ions, such as K^+^, Rb^+^, Cs^+^, and NH_4_^+^, stabilize the helix of κ-carr and promote the gelation. κ-carr gelation occurs in pure NaI, or in mixtures of NaI and CsI with low content of CsI, by reversible association of helical dimers or by formation of super-helical rods [266]. The chain stiffness is considerably influenced by the conformation which is adopted by κ-carr due to the electrostatic and intermolecular interactions in well-established conditions [267].

κ-carr/PAAm DN hydrogels with mechanical performances and good stability in water were reported [268]. The first network was achieved by sol-gel transition of κ-carr at room temperature due to the coiled-coil junctions. The second network was obtained by photopolymerization of acrylamide, then the DN was incubated in presence of multivalent ions (Zr^4+^, Al^3+^, Fe^3+^, Sm^3+^) when the polysaccharide forms coordination complexes. Zr^4+^ ion presents high charge and small radius of 0.72 Å (it has an empty 5s orbit and the 4p orbit is partially empty), forming strong coordination bonds with κ-carr; thus, the tensile breaking stress of tough DN hydrogels was of 1.5–3.2 MPa, Young modulus of 0.2 MPa–2.2 MPa, breaking strain of 300–2200% and tearing fracture energy of 0.4 kJ/m^2^–18.5 kJ/m^2^.

Interfacial polyionic complexation of *oppositely charged polysaccharide pairs* (i.e., positively charged CS with negatively charged κ-carr, alginate, or gellan gum) was used to produce continuous interfacial fibers with strong mechanical properties [269]. These bioinspired hydrogels presented multiscale hierarchy (parallel or perpendicular alignment), ability to encapsulate NPs, small molecules or to form composites with other macromolecules [270,271].

*Xanthan gum* (XG) is an anionic polysaccharide characterized by dimensional and structural polydispersity [272]. The primary structure of XG is mainly composed by 5 repeating units (Figure 16), with a segment of two β-(1-4)-d-glucose units linked at positions 1 and 4 (as in cellulose), at each second unit it is attached a trisaccharide side chain at C-3, consisting of two mannose residues and a glucuronic acid residue between them. Mannose residues may present a C-6 acetyl group and the terminal mannose residue attaches a pyruvate group between C-4 and C-6. The fermentation conditions determine the content of pyruvate and acetyl groups, influencing the stability of the xanthan conformation (helical ordered structure at room temperature).

Inspired from enzyme-resistant dietary fibers, an injectable anti-digestive hydrogel was obtained by photopolymerization of glycidyl methacrylate-modified XG [273], with porosity, swelling ratio, and stiffness depending on polymer concentration. Ca^2+^ ions trigger the hydrogel shrinkage and help to remove it timeously. The hydrogel was tested as gut repair material, showing efficiency in closing the gastrointestinal fistula. It presented a high viscosity at rest, contributing to its tight adhesion to the surrounding tissues and thus preventing the intestinal juice loss.

*Dextran* (Dex) is a polysaccharide of microbial origin with branched poly(α-d-glucoside) structure, having predominantly α-(1,6)glycosidic bonds (Figure 17).

Inspired by the adhesion and curing ability of mussel glue, an adhesive hydrogel formed in situ via double dynamic bonds (catechol-Fe and Schiff’s base bonds) was obtained using catechol-modified ε-poly-l-lysine and oxidized dextran (OxDex). The catechol and aldehyde groups contribute to an improvement of the tissue adhesion through multiple non-covalent and covalent interactions. The dynamic bonds give unique features to the hydrogels: reversible breakage and reformation, dissolution on demand, repeatable adhesiveness, injectability and biocompatibility, and adhesive and mechanical strength appropriate for wound closure and healing [274]. OxDex and gelatin were used to design an extrudable hydrogel for 3D bioprinting applications. Due to temperature sensitive properties induced by the phase separation phenomenon, the hydrogel presented tunable gelation time and under physiological pH conditions [275].

### 3.5. Collagen

Collagen is a mixture of proteins and peptides collected from skins, bones, or connective tissues of animals, being frequently used for designing hydrogels, especially for tissue engineering. It has a high ability for self-aggregating by hydrogen bonds, forming gels with different structures and functionalities, able to mimic ECM structure and to improve the biocompatibility (adhesion). Collagen molecule is composed by at least partly of triple helices of amino acids twisting in a fibrous structure, found in almost all tissues. Collagen type I (Figure 18) is abundant in the skin, tendon, and bone, where the types III and V exist in smaller content. The type II collagen is found mainly in cartilages [276].

The collagen-based materials with complex structures able to rebuild different shaped products, such as membranes, sponges, threads for surgical or dental purposes, and cell culture matrices, are obtained by different physical (electrospinning, molding, or additive manufacturing) or chemical procedures. Collagen based bio-inks can be formed into 2D or 3D matrix, in solid or liquid form, combined with living cells, as it is required in pharmaceutical and medical applications (artificial tissues: heart valves, skin, cartilage, breast reconstruction, vocal, and spinal cord) [98,102,120,265,277,278].

### 3.6. Gelatin

Gelatin (Figure 19) is a readily available water-soluble protein which is obtained by partial hydrolysis of collagen, maintaining the key collagen sequences that contribute to biomaterial-cell interactions [279].

It undergoes a thermoreversible gelation, presenting gel-like behavior bellow the gelation temperature (located between 20 °C and 30 °C) and liquid-like properties at higher temperatures. Ampholytic swelling behavior in salted solutions is influenced by pH value, influencing the elastic properties of the gelatin gels [280]. Various crosslinking strategies (including physical, chemical, or enzymatical methods [31,32,281,282,283], or the creation of IPNs with polysaccharides [284,285,286] were considered to realize stable hydrogels at physiological temperature and to improve the mechanical properties.

Gelatin based hydrogels are used in many applications, from ingredient of food preparation to innovative materials and biomaterials. Thermoreversible IPNs were prepared by addition of genipin or glutaraldehide to N-isopropylacrylamide/gelatin mixtures [287]. A rapid swelling/deswelling occurs when the temperature was cycled from 25 °C to 37 °C, regardless the content of gelatin and crosslinking agent. Genipin (Figure 20) is obtained from gardenia fruits and it is used for the fabrication of blue pigments for food applications [288]. In the last decades genipin was very often used as non-toxic crosslinker [50,283].

In order to avoid the cytotoxicity of cross-linking molecules or release of cytotoxic degradation products, many efforts were concentred on non-toxic gelatin derivatives (as for example methacryloyl gelatin, GMA [289,290]) or the use of enzymes (e.g., the added microbial transglutaminase can link covalently the glutamine and lysine residues from gelatin [291]). Biocompatible, bio-inspired inverse opal scaffolds with self-healing characteristics were developed by using GMA, BSA hydrogel filler (crosslinked with glutaraldehide) and enzyme additives (glucose oxidase and catalase) [292]. The assembling and healing of different elements in this hybrid hydrogel create a structural color, whereas the reversible covalent attachments of glutaraldehyde to lysine residues of BSA and enzyme additives ensure the self-healing behavior. An interesting method, called zero length crosslinking, was reported by Campiglio et al. [293], which allows covalent bond formation in the presence of a crosslinker that is further removed from the gelatin network. Gelatin hydrogels with adhesive and tunable elastic modulus were obtained via click chemistry reactions of tetrazine and norbornene [294]. The functionalization and dynamic cross-linking were performed with norbornene-modified gelatin that allows dynamic stiffening and control over stress relaxation through dimerization, hydrazone bonding, or boronate ester bonding.

The printability of gelatin-alginate bio-inks of different compositions was evaluated through the dynamic moduli (G′ and G″) and loss tangent (G″/G′). An increase in G′ or G″ was correlated with the rise of the pressure required to extrude the bio-inks. For lower values of loss tangent the structural integrity was conserved and for higher tanδ an increased extrusion uniformity was observed. It was found that the loss tangent values between 0.25 and 0.45 correspond to the optimal region where there is the best compromise between structural integrity and extrusion uniformity for gelatin-alginate hydrogels [295].

## 4. Concluding Remarks

Inspired from the biological systems, researchers try to mimic the natural processes by using appropriate approaches for the hydrogel design, obtaining hydrogels with structural and functional diversity guaranteeing the requirements of targeted applications (networks with tunable characteristics that can regulate their functions in the targeted applications) [296,297].

Various materials with complex structures at different scales are reproduced in laboratory by using adequate techniques in order to reproduce the functions existing in living organisms. Bioinspired procedures are remarkable tools for the optimization of drug delivery systems and health monitors. A peculiar interest is focused on bioinspired or biomimetic self-repairing materials, using the mechanisms found in nature to design of new or improved materials with self-repair function. Hence, the present review pays more attention to bioinspired approaches and biomacromolecules used recently for hydrogel formulation in order to solve the complex requirements of applications in biomedical and beyond medical area (Figure 1).

## 5. Future Perspective

A wide variety of inventive hydrogels reported recently in the literature were inspired by structural and functional principles that govern behavior in the living world. Many of the approaches used to design multifunctional networks provide forward-looking ideas for new structural design of advanced materials. An intensive development of the hydrogels in the future supposes complex activities, involving deep research and advanced concepts in physico-chemistry of polymers, materials science, biology, medicine, pharmacy, and biotechnology. Smart nanomaterials with unique characteristics, as promising theranostic tools for both diagnosis and therapy, will revolutionize medicine [298].

Hydrogels submitted to prolonged loads undergo fatigue, with changes of properties and nucleation/growth of cracks. Reversible bonds are able to prevent the fatigue damages, whereas sacrificial bonds act as tougheners [299]. Specific to gels is poroelastic fatigue evidenced by stress-relaxation in conditions of prolonged static stretch. Suitable strategies to design hydrogels of high endurance are challenges for the next period.

An original approach to test the cellular feature is viscoelasticity of ECM that allow the differentiation of healthy and diseased tissues, such as cancer progression and its therapy [300]. Due to the changes in composition and structure of ECM in the case of tumors as compared to healthy tissue, ECM of the tumor becomes richer, denser, and stiffer, acting as a barrier that impedes the diffusion of oxygen, nutrients, and metabolites. The knowledge of the stiffness for wounds and inflammatory tissues is beneficial in developing new therapies during the cancer progression.

The properties of hydrogels for a targeted application can be improved through synergistic combination of different biomolecules with synthetic polymers and fabricating new structures using the traditional or innovative crosslinking techniques. The purity of materials, side reactions during the network synthesis, and chain entanglements in a specific environment should be taken into account to improve the synthetic methods for hydrogel fabrication.

The new approaches try to overcome some deficiencies of many hydrogels concerning mechanical robustness, slow or delayed response times to external stimuli, loading and release of therapeutic agents, etc.—or to include new properties and functionalities—biocompatibility, biodegradability, immunological compatibility, prolonged therapeutics, easy handling, and use, etc. [301,302]. Alternatives for overcoming the actual limitations are also given by the nature principle applied in orthogonal crosslinking or self-assembling methods, multi-component and hybrid networks, stimuli-responsive hydrogels and nanogels, 3D printed hydrogels, and electrospun nanofibers [30,31,32,33]. The final costs for biomaterials and the environmental protection cannot be neglected when trying to manufacture them on a large scale.

Despite of the high number and variety of existing hydrogels, the translation of biomaterial candidates into the clinical setting is rarely met and this remains as a challenge for the upcoming period. Clinical practices and sustainability in close connection with the design of new hydrogels will allow an important advance of biomaterials as platforms for life-science applications.

## Data Availability

Not applicable.

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
