# Peer review of "Bioinspired Hydrogels as Platforms for Life-Science Applications: Challenges and Opportunities"

_polymers, 2022, doi:10.3390/polym14122365_

Round 1

Reviewer 1 Report

Comments attached

Author Response

I am grateful to Reviewer 1 for the careful analysis and constructive comments that helped me improve the content of this review.

Reviewer 2 Report

Summary 

The manuscript entitled "Bioinspired Hydrogels with Tunable Properties: Challenges and Opportunities" by Maria Bercea is directed toward reviewing recent advances in the field of bioinspired hydrogels that can serve as platforms for life-science applications. The authors showed and clarified different issues related to the revised systems, and several types of approaches to using them have been discussed.

General comments 

In general, the work is accurate and clearly presented, and the given outcomes are of interest to the readers of Polymers. 

The work contains an overview of the applications of biomimetic hydrogels for biomedical applications. This is of interest by itself; moreover, understanding the properties of these materials is vital for designing effective systems and therapeutic agents. It is a timely topic and of interest to Polymers readers, as the scientific community is strongly using biomimetic biomaterials for these applications. A complete understanding of the use of hydrogels is lacking, and this review attempts to address this topic.      

The article's English grammar and style are correct, but it should be reviewed on a few points. Thus, I believe that the text needs some technical adjustments to be published. Therefore, I recommend that this manuscript can be published in Polymers after Minor Revision. 

Specific comments

Going into details on the specific issues, here some comments are reported:

- The Title should be modified indeed, including" Tunable Properties" in the Title stressed this point too much, while this is only part of that manuscript.

- The author wrote: "In particular, smart networks able to respond to physical, chemical and biological stimuli gained a high attention for a wide range of applications: tissue engineering [4], bone regeneration [5], controlled-release drug delivery vehicles [6], wound healing [7], soft robotics [8], intelligent electronics and artificial intelligence [9], actuators [10-12], stretched electronic devices [13], hygiene products [14,15], contact lens [16], cosmetics [17], food nutrition and health, food safety and food engineering and processing [18], advanced wastewater treatment [19], catalysis [20-22], etc.".

Here one of the most essential and forefront biomedical application areas is missed. Please add "biosensing" to the list, including this brand-new impactful article as a reference for it [https://doi.org/10.1038/s41427-022-00365-9].

- Dr. Bercea claimed that ‘’’and a brief outlook on the actual trends and future directions is also presented.'". I believe this is too brief, and a paper like this deserves a short separate section about Future Perspective

- Frankly, there is no proportion between text length and the number of Figures. Please add at least another Figure to the manuscript.

Conclusion

The topic of this manuscript falls within the scope of Polymers. I like the structure and content reported in this paper; moreover, the manuscript includes an in-depth discussion of the investigated materials' chemical, structural and applicative descriptions. Anyway, I think the manuscript needs a few improvements. I believe the article is of sufficient quality to meet the Polymers publication standards after a Minor Revision.

Author Response

I am grateful to Reviewer 2 for the careful analysis and constructive comments that helped me improve the content of this review.
